# Time Distribution of Strong Seismic Events in the Fore-Sudetic Monocline in Context of Signals Registered by Water-Tube Gauges in Książ Geodynamic Laboratory

**DOI:** 10.3390/s21051603

**Published:** 2021-02-25

**Authors:** Marek Kaczorowski, Damian Kasza, Ryszard Zdunek, Roman Wronowski

**Affiliations:** 1Space Research Centre, Polish Academy of Sciences, Bartycka 18A St., 00-716 Warsaw, Poland; marekk@cbk.waw.pl (M.K.); rysiek@cbk.waw.pl (R.Z.); roman@cbk.waw.pl (R.W.); 2Faculty of Geoengineering, Mining and Geology, Wrocław University of Science and Technology, Wyb. Wyspianskiego 27 St., 50-370 Wrocław, Poland

**Keywords:** geodynamics, seismic activity, tectonic activity, water-tube tiltmeters, Świebodzice Depression, Fore-Sudetic Monocline

## Abstract

Changes in the stress field in Świebodzice Depression (ŚU) unit area are the reason of complex kinematics of the rock blocks consisting of rotations and horizontal/vertical displacements. The measurement system of the Geodynamic Laboratory in Książ, associated with rock blocks which are separated by faults, is a natural detector of tectonic activity. Installed in laboratory long water-tube gauges allowing to determine the functions of tectonic activity—TAF, and their derivatives. A comparison of the TAF with the seismic activity of the Fore-Sudetic Monocline showed that the strong seismic shocks (magnitude ≥3.6) occur in the Monocline only during defined and repeatable phases of the kinematic activity of the ŚU. Observed concordance proves the thesis of the existence of a large-scale, and largely homogeneous field of tectonic forces which, at the same time, cover the ŚU and the Fore-Sudetic Monocline units. The results of comparison between seismic events temporal distribution and phases of tectonic activity of the ŚU orogen indicate existence of the time relation between function of derivative of the tectonic activity (TAF) and seismic events.

## 1. Assumption about a Homogeneous Large-Scale Field of Tectonic Forces in the Central Europe

Carrying out research on the time dependences of the seismic activity in the Fore-Sudetic Monocline (FSM) and the tectonic activity of the Świebodzice Depression unit (ŚU) are justify only in the case of correctness of the thesis about the existence of a large-scale, well-approximated homogeneous field of tectonic forces, which simultaneously encompass the two geological units FSM and ŚU. The credibility of this thesis is confirmed by the results of long lasting observations of the Earths’ crust motions performed on hundreds of measurement stations by space and satellite tech-niques (i.e., GNSS, SLR, DORIS, VLBA) [1,2,3,4,5,6,7,8,9] which were used to develop the global models of tectonic plate velocities, e.g., ITRF2008, ITRF2014 or DTRF2014 (Figure 1) [10,11,12,13,14].The measurement series from numerous stations and different measure technic, located on the surface of the main tectonic elements of the Earth’s crust, such as plates and continental blocks, show uniform velocity field in good approximation. This observation is repeated on the most part of the Earth surface, except for the areas of rifts, subduction zones and other oddities, e.g., areas of continental plates collisions. Homogeneous velocity fields indicate the passive role of continental plates the motion of which is produced by fields of tectonic forces external to the plates.

This situation can be found in the area of Central Europe, where there are no above-mentioned tectonic curiosities. In the Polish Lowlands, differences between GNSS stations velocities do not exceed a single percent of the average value of the horizontal component of the Euro-Plate velocity, i.e., about 30 mm/year [16]. In the area of the shallow crystalline substrate such as the Sudetic and Fore-Sudetic Blocks [17,18], as well as the Czech Massif [19,20], an increase in the velocity differences between GNSS stations in comparison to differences between the velocity of stations in the Polish Lowlands is observed. This effect highlights the process of averaging the velocity between stations caused by a thick layer of sedimentary rocks covering the crystalline substrate in the Polish Lowlands. According to the principles of mechanics, a homogeneous field of velocity of the body is associated with a homogeneous field of forces. This field of tectonic forces ought to be collinear to the velocity field. Therefore, in the area of the Czech Massif, the Sudetic Mountains (Świebodzice Depression), Fore-Sudetic Monocline and Polish Lowlands, we should expect a homogeneous field of tectonic forces in large-scale.

Additional confirmation of the thesis about a large-scale homogeneous field of tectonic forces is provided by the reaction of the Sudetic Marginal Fault (SMF) which is affected by tectonic movements generated by this field of tectonic forces [21,22,23,24]. The TM-71 crack gauges installed on opposite wings of SMF register mainly vertical motions [24], with simultaneous lack of observations of the horizontal displacements. These results confirm that direction of the large-scale field of tectonic forces is perpendicular to the strike of the SMF. In large sections the route of SMF is normal in approximation to the average direction of velocity vectors of the GNSS stations as well as normal to the expected direction of the tectonic field of forces (Figure 1).

The assumption of the thesis about a homogeneous large-scale field of tectonic forces suggests that observations of the ongoing process of deformation of the Świebodzice Depression unit, provide information about current state of deformation of the Fore-Sudetic Monocline.

## 2. Natural Conditions for Registration of Geophysical Signals in the Książ Geodynamic Laboratory

Geodynamic Laboratory (GL) of Space Research Centre PAS (SRC PAS) was established in underground tunnels built in the ŚU orogen (Figure 2). The ŚU is specially intersected on account of dense network of faults, which separate single blocks as well as the neighboring geological units such as Intra-Sudetic Basin (the Struga Fault in the west), Sowie Range Gneissic Block (the Szczawienko Fault in the south), Fore-Sudetic Block (the Sudetic Marginal Fault in the east) and Kaczawa Metamorphic Complex in the north with complex system of faults [25].

The morphology of the terrain, with numerous outcrops in which the fault lines are well visible, is especially favourable for the investigations and monitoring signals of the recent tectonic activity. Dense network of faults arose during about 400 million years long geological history of massif. Rich system of faults guarantees large freedom of motions of rocky blocks of ŚU and aseismic dissipation of tectonic stresses.

Installed in such conditions measure gauges such as WTs, SRDN-3 radon probes are suitable for registration of the subtle tectonic signals of kinematic as well as of geochemical character (changes of radon concentration). Specially, suitable for these purposes are WTs because of their large size which make possible measuring effects of tilting and vertical motions of foundation below the tubes of tiltmeters.

The Świebodzice Depression rocks are built out of strongly bound conglomerates from the Carboniferous-Devonian Epoch, which consist of metamorphic gneiss rocks from the nearby Sowie Range [26,27].

Thickness of Świebodzice Depression conglomerates is eveluated to 10 km or more [28,29,30] what satisfy excellent propagation of a wide range and geodynamic signals: seismic, tidal, the Earth’s free vibrations as well as tectonic. High level of these signals is documented by seismographs of the Institute of Geophysics of the Polish Academy of Sciences and by tiltmeters and other instruments of GL of PAS. The GL in Książ is the only research unit in Poland equipped with the set of instruments enabling to tectonic signals registration with nanometric resolution. These capabilities allow to observe the tectonic deformations of Książ massif in real time [25,31,32,33,34].

## 3. The Tectonic Signals Registered by Instruments Installed in Książ Underground Laboratory

There are three kinds of signals which are registered and investigated in GL. The first kind of signals are deformations of massif in aspects of tilting and vertical motions (measured by horizontal pendulums and water-tube tiltmeters) [32,35,36,37,38], the second kind of signals is horizontal displacements of the blocks of main southern fault (measured by two GNNS stations installed on opposite blocks of fault) [34] and the third signal of geochemical origin i.e., changes in radon concentrations measured by AlphaGUARD and SRDN-3 probes [39].

The first indicators of existence in the ŚU massif tectonic signals were obtained with help of the quartz horizontal pendulums (HPs). The HPs that work in GL since the 1970s registered numerous tectonic events occurred in the ŚU (Figure 3A) [40]. Over forty years long series of HPs observations were basis for determination of the temporal and amplitude characteristics of tectonic phenomena. Observed tilting effects reach extremely hundred to thousand miliarcseconds (MAS) and last dozen or so days.

Signals of similar characteristics in amplitude and time domain were registered by two water-tube tiltmeters (WTs) installed at the GL at the beginning of this century [35]. Different class tiltmeters—HPs and WT with different method or measurements confirmed existence of strong tectonic signals in Świebodzice Depression massif (Figure 3B).

Because of HPs properties such as small size of the measuring base as well as onehundred time lower sensitivity than WTs, the HPs provide much less information about the current state of tectonic activity of the ŚU than WTs. Additionally, the WTs register vertical motions of rock blocks that are in the measurement range of the WTs. Because of the numerous faults across the line of the WTs tubes (Figure 4) the effects of vertical motions of rock blocks provide in general stronger signals than effects of tilting of foundation.

### 3.1. Tectonic Activity Function (TAF) of the Świebodzice Depression

Time and amplitude variations of tectonic activity of the Świebodzice Depression have been described by the Tectonic Activity Function [39,41]. The TAF is an empirical function determined on the basis of measurements carried out with instruments that have a small, linear and well-defined drift, high sensitivity to orogen deformations as well as extensive and large-scale measure base. The above-mentioned conditions are well satisfied by suite of two perpendicular WTs. The length of the tube (WT1-2) in azimuth 58.6° is 65.24 m and in azimuth 148.6° (WT3-4), length of the tube is 83.51 m respectively. Furthermore, the instrumental drift of WTs is generated mainly by water evaporation from hydrodynamic system of gauges. Evaporation drift effect is linear in approximation and the sensitivity of the measure system is close to several nanometers [31,42].

The size of WTs measure base is much larger than the average distances between separated by faults single rock blocks (Figure 4). Large size of the registration system of WTs guarantees its high representativeness in the assessment of the momentary tectonic activity of the orogen as well as provide information in the aspect of vertical motions and tilting of the foundation [25,32,36,37,38].

The tectonic signals are usually one hundred times stronger than the tidal signals (Figure 3). During the largest tectonic events tectonic signals exceed several hundred times the amplitude of the tidal signals [38].

The geodynamic signals registered by WTs are related to the orogen to which the base lenses of the four WTs interferometric gauges are rigidly connected.

The tectonic signals which are observed by WTs are produced by two types of phenomena: the tilting effects of the whole massif and the vertical motions of the separated by faults rock blocks under the tubes of WTs (Figure 4). Therefore, the resultant tectonic signal consist of two components is given by the following formula:(1)$$S\left(a,r\left(l\right)\right)=L*\cos\left(a\right)+\int_{l=0}^{l=L}r\left(l\right)dl$$
where *L* is the length of the WT tube and *a* is an angle of tilting of the orogen as a whole.

Function *r*(*l*) describes the vertical deformations of the orogen under the tubes of the WT [31]. The signals generated by the vertical motions is given by the integral of *r*(*l*) function counted along the WT tubes. The WT’s system creates non-linear superposition of both types of signals, the mutual proportions of which change in the time.

TAF function is calculated based on raw series of observations *S*(*a*,*r*(*l*)) in several steps. At the beginning, from *S*(*a*,*r*(*l*)) series are removed jumps and discontinuities produced by the instrumental effects or by the strong seismic events. Then, the high-frequency signals are filtered and effect of water evaporation is eliminated. Next, the ETERNA 3.4 package is used for reduction of tidal signals and determination of the tectonic signals [43]. In the final step of elaboration, the TAF time series are justified with polynomial splines of second order to obtain suitable smooth function. Smoothing of the function makes possible to determine the derivative of TAF functions i.e., velocity function of the orogens’ deformation—VTAF. Four derivative functions VTAF are calculated on the basis of measurements of four measurement channels of WTs. The VTAF play important role in further discussion presented in this paper.

### 3.2. Exsamples of the Strongest Seismic Events in the FSM and Derivatives of Tectonic Activity Functions of the ŚU in the Years 2013–2017

Over a dozen years long series of TAF and their derivatives VTAF enabled to perform comparative studies of time relations between variations of kinematic activity of the ŚU and time distribution of strong (magnitude ≥3.6) seismic events in the FSM. In this paper we take to the discussion only earthquakes of magnitudes 3.6 or larger (the strongest is 4.9). Strong energy events occurred less than 20 times in the year while weaker earthquakes <3.6 [Mag] occurred several hundred times per year.

The strongest seismic events in context of derivatives of tectonic activity functions (VTAF) were shown on plots in Figure 5. The Earthquakes take place at the distinctive phases of the orogen deformations described by time and amplitude rulers (see Section 3.3).

In the Figure 5 it is well visible that strong events occurred when velocity of deformations of massif decrease to zero and plots of derivatives changes their signs. Very specific moments on plots from Figure 5, when earthquakes occurred, confirm existence time relations between variations of kinematic activity of the ŚU and time distribution of the strong seismic events in the FSM.

### 3.3. Definitions of the Rules of Deformation of the Świebodzice Depression Which Are Preciding the Seismic Activity of the Fore-Sudetic Monocline

Foundation for the further discussion presented in this paper is the assumptions presented in Section 1 i.e., existence of the homogeneous large-scale field of tectonic forces covering simultaneously the ŚU and the FSM geological units. Correctness of this assumption allows to conclude that results of analysis of the process of deformation of the ŚU provide information about the state of deformation of the FSM.

Discussion of the phases of kinematic activity of the ŚU revealed that the Książ massif reaches states corresponding to the periods of increased seismic hazard in the FSM according to defined and repeating time and amplitude rules. This observation opens the possibility of discussion of the process of deformation of the ŚU massif which precede periods of seismic activity in the FSM.

The recognition of the principles of deformations of the ŚU massif make possible application the time and amplitude rules for evaluation of the level of the seismic hazard in the FSM. Each of the time and amplitude rules consist of the five precedents. Definitions of precedence were shown in graphical form in Figure 6.

The time precedents specify the interval (in hours) between the last zero crossing by the VTAF and seismic event (see time precedents—Table 1). The values of the time limiters which were applied for definition of time precedents T, were selected empirically (Table 1).

The amplitude precedents A are defined by values of the VTAF at the moment of the seismic event for four channels of the WTs. The magnitudes of the limiters applied for definitions of precedents were selected empirically and their values result from the multiplier 10^3^ (Table 2) which was used to move the further discussion in to the space of integer numbers.

Symmetry and double symmetry of signals (Table 2 No. 3 and 4) concern measurement channels from opposite ends of the same tube- symmetry of signals indicate tilting effects. Symmetry of signals from different tubes are not discuss.

## 4. Fundamental Principles Which Must Be Satisfy before Earthquakes and Precedents Registered during the Seismic Events

A graphical presentation of precedents which take part in the FSM during seismic events (magnitude ≥ 3.6) in period 2013–2017 were shown in Figure 7A, Figure 8A, Figure 9A, Figure 10A and Figure 11A (the time precedents—T) and in Figure 7B, Figure 8B, Figure 9B, Figure 10B and Figure 11B (the amplitude precedents—A).

Elaboration of the empirical results presented in Figure 7A, Figure 8A, Figure 9A, Figure 10A and Figure 11A and in Figure 7B, Figure 8B, Figure 9B, Figure 10B and Figure 11B revealed fundamental principles in time and in amplitude domains which must be satisfy before seismic event: VTAF of four measure channels must pass zero before earthquake,Last zero pass of VTAF ought to be executed no late than ca. hundred hours before seismic event,At the moment of the seismic shock at least one channel—VTAF value, ought to be focused around zero, i.e., in the range of −22,000 to 22,000 μm/h.

An exception to the last principle is a special case of double symmetry of VTAF channels. Double symmetry o values of VTAF means the symmetry between opposite channels of WTs i.e., 01 to 02 and 03 to 04 (see Figure 9B on event No. 7).

It was observed that almost all of the seismic shocks (magnitude ≥ 3.6), occurred during the phases of low velocity of deformation of the ŚU i.e., small absolute values of VTAF (VTAF in the range of −22,000 to 22,000 μm/h) or symmetry of VTAF values (Figure 6; Table 3; Figure 7B, Figure 8B, Figure 9B, Figure 10B and Figure 11B).

This regularity is well visible in the Figure 7B, Figure 8B, Figure 9B, Figure 10B and Figure 11B. Values of VTAF are close to zero at the moment of the seismic shock. Improvement of this regularity follow increase of magnitude of seismic shock.

In further text was presented short description of graphical presentation of time and amplitude precedents from Figure 7, Figure 8, Figure 9, Figure 10 and Figure 11 which accompanied with successive seismic events (3.6 or larger magnitude) in the years 2013–2017:Year 2013 Figure 7A,B

This year, values of VTAF in time of seismic events were contained in the range from −22,000 to 22,000 µm/h during four (4) events. Event no. 2 occurred in the moment of symmetry of very large distant channels no. 3 and no. 4.

For event (no. 1) zero crossing of all the VTAF happened almost at the moment of earthquake (very rare case) (Figure 7A and Table 3). For very strong seismic events (magnitude 4.6, no. 3) the time and amplitude precedents were perfectly satisfied. In 2013 double concentration of derivatives values occurred for strong events (magnitude 4.1, no. 9). Cases of three-fold concentration close to zero and one channel distant from zero happened for events no.6 and no.11. Case with one channel close to zero and three-fold concentration of the VTAF happened for event (magnitude 3.8, no. 8). All earthquakes in 2013 take place less than ca. 50 h after the last zero crossing by the VTAF. Two events no. 8 and no. 10 occurred with large overtake of the variation of the VTAF sign on channel no. 03. See in the Table 3 zero pass of channel no. 4 Figure 7A.

Year 2014 Figure 8A,B

In 2014 double concentration of the VTAF values occurred two times for strong events (magnitude 4, no. 2) and very close (magnitude 4.6, no. 4). Three-fold concentration of VTAF with one channel close to zero happened for strong event (magnitude 4.1, no. 1). This year double symmetry of derivative values took place three times during low energy shocks (magnitude 3.6, no. 5, magnitude 3.8, no. 6 and magnitude 3.7, no. 7). Invent no. 3 (magnitude 3.8) occurred with closed to zero double concentration and weak symmetry of channels 01 and 02 with very distant values of VTAF channels no. 01 and no. 02. Invent no.3 is a case of weak double symmetry. All earthquakes in 2014 take place less than ca. 40 h after the last zero crossing by the VTAF. Zero crossing of the channel no. 01 as well as channel no. 04 largely overtake zero crossing of other channels during three events no. 3, no. 7 and no. 9 Figure 8A.

Year 2015 Figure 9A,B

Event no. 1 (magnitude 3.9) happened during double symmetry of VTAF values and at the moment of almost simultaneous zero crossing of all the VTAF. Particular case of double symmetry of VTAF occurred second time during strong event (magnitude 4.0, no. 7). This earthquake occurred despite of the fact that one pair of derivatives were totally outside the range of −22,000 to 22,000 µm/h and the second pare were outside of the range −15,000 to 15,000 µm/h. The last case indicates on important role of symmetry which turned out orogen to the state strongly seismogenic. In 2015 double concentration of VTAF values occurred for two strong events (magnitude 4.4, no. 4) and (magnitude 4.1, no. 5). Event no. 4 happened with channel 01 closed to zero, concentration of moments of zero crossing of channels 03 and 04 and large overtake of zero crossing by channel 02. Except events (no. 6, no. 7 and no. 8) values of other measure channels of VTAF were in range of −20,000 to 20,000 µm/h, during earthquakes. In 2015, one seismic shock (magnitude 3.7, no. 8) had a weakly defined amplitude precedent (Table 3). Except for the particular invent no. 11 other events take place less than ca. 50 h after the last VTAF channel zero crossing in 2015. During this event channel no. 3 overtake other channels zero crossing which obtained zero almost at the same moment and in surprise large distant before earthquake (ca. 120 h). It suggests that weak event (3.7 Mag) no. 11 was anthropogenic.

Year 2016 Figure 10A,B

During ten seismic events which occurred in 2016 values of VTAF on the measure channels (no. 1, 3, 4, 5, 6, 7, 10, 11, 14 and 15), were concentrated in the range −20,000 to 20,000 µm/h.

For event (no. 9) value of VTAF during earthquakes were slightly large i.e., in the range −22,000 to 22,000 µm/h. For events no. 4, 5, 7, 10 and 11 there happened three-fold concentration of moments of zero crossing of VTAF which overtake earthquake ca. 120 h and with single channel 03 which overtake all these events by few hundred hours. Similar situation take place for events no. 3, 13 and 14 but with ahead channels 01 or 02. Double symmetry of the values of VTAF occurred at the moments of five events (no. 1, 2, 9, 10, 11, 12)—Figure 10B. As in the previous years all earthquakes from 2016 take place less than ca. 50 h after the last zero crossing by the VTAF with exception of invent no. 1. During these invent four channels of VTAF simultaneously crossed zero ca. 100 h before earthquake. Event no. 1 is a rarely case because the principle of minimum one channel zero crossing 100 h before earthquake was satisfy at the last moment (see Time precedents, Table 3). Event no. 1 occurred at the moment when values of VTAFs reached a double symmetry. This year one low energy seismic shock (magnitude 3.7, no. 13) had a weakly defined time and amplitude precedents.

Year 2017 Figure 11A,B

In 2017 for eleven (11) events no. 4, 6, 7, 8, 9, 10, 11, 12 and 13, before seismic events were observed low velocities of deformations of Książ massif i.e., all VTAF values were in the range −20,000 to 20,000 µm/h. For events no. 1, 2 and 5 values of VTAF were in the range of values −22,000 to 22,000 µm/h. For strong event (magnitude 4.3, no. 3) values of two VTAF channels 03 and 04 only were in the range −20,000 to 20,000 µm/h while channel 01 was distant −27,000 µm/h and 02 was distant −50,000 µm/h. Five hours before event no. 3 values of VTAF on channel 04 obtained zero which satisfy principle of four channels zero passing by VTAF before event (see Time precedents, Table 3). During event no.1 (magnitude 4.0) four VTAF channels were in the range of −22,000 to 22,000 µm/h and 4 VTAF channels zero passing several hours before earthquake. As in the previous years all earthquakes from 2017 take place less than ca. 50 h after the last zero crossing by the VTAF.

## 5. The Seismic Events in Contexts of VTAFs and Precedents Distribution

In the years 2013–2017, 57 strong seismic events of magnitudes equal or greater than 3.6 Mag occurred in the FSM. The results of comparative studies on the kinematic activity of the ŚU with distribution of seismic events were presented in Table 3.

Table 3 contain the values of VTAF at the moments of seismic events, the time delays of the moments of last passage of the VTAF through zero before the seismic events and individual names of time and amplitude precedents corresponding to seismic events.

Moreover, the headings of table refer to the dates and magnitudes of events. For strong seismic events (magnitude ≥ 3.6) happened in the years 2013–2017, almost all the time precedents as well as the amplitude precedents were implemented by one of the time and one of the amplitude precedent defined on the Figure 2, Table 1 and Table 2. The seismic data applied in this paper were taken from EMSC catalog [44].

## 6. Discussion

Defined time and amplitude precedents well represent almost all cases of the tectonic situations which occurred during the earthquakes (see Figure 7, and Table 1, Table 2 and Table 3). For one event only, there was lack of definition for low energy shock (3.7 > Mag)—from 15 October 2016 (Table 3).

The results of elaboration of period 2013–2017 show that increase of energy of shocks cause improve determination of precedents. For weaker shocks determination of precedents is significantly worse. This observation indicates that low-energy shocks are probably of an anthropogenic origin.

In the following Table 4 was showed combination of precedents from Table 3 in relation to seismic events energy.

In the years 2013 to 2017 in FSM happened 57 earthquakes of energy (Mag ≥ 3.6) [44]. 17 cases of seismic events have had magnitude 3.8/3.9. Six of these events took place in 2013.

For seismic events of energy less than 4.1 Mag the total sum of the seismic events were approximately similar in the analyzed period (2013–2017). Fast decreasing of the sum of events for energy greater than 4.1 Mag was observed. The strongest event 4.8 Mag happened in 2017. Significant increase of the number of seismic events in the years 2016 and 2017 was observed. Combination from the Table 4 appear that the amplitude precedent which is the most frequently invocated during the earthquakes is A1.

In the years 2013 to 2017 the A1 precedent was invocated 31 times (see Table 2 and Figure 7, Figure 8, Figure 9, Figure 10 and Figure 11) (four-fold concentration of the VTAF close to zero). The second amplitude precedent which was invocated 12 times is A2a (see Table 2 and Table 4). The other amplitude precedents were applied much seldom for definition of tectonic situation in the time of the earthquakes from the years 2013 to 2017.

This observation indicates that effect of slowing-down of kinematic activity of the massif play important role in the process of the increasing of the orogen susceptibility to destruction thus increase possibility of seismic events occurrence. The concentration close to zero of the amplitude precedents at the moments of shocks confirm this thesis Figure 7B, Figure 8B, Figure 9B, Figure 10B and Figure 11B.

From the Table 4 result that time precedent which was the most frequently invocated during seismic events is T2. The precedent T2 was call 23 times while precedent T3 was call 11 times and precedent T1 was invocated 10 times only. These results are surprising because of very strong condition of the precedent T1 (see definitions from Table 1 and Figure 7A, Figure 8A, Figure 9A, Figure 10A and Figure 11A. The third position of the time precedent T1 is related to the large number of invocations of T2 and T3 precedents for shocks of relatively low energy (<3.9 Mag; see Table 4). This example shows that fulfilment of the time precedents is not so strong critical for seismic events triggering as amplitude precedents. Nevertheless, on the basis of observations we known that four measure channels of VTAF must pass zero less than 600 h before earthquake and one of them less than 100 h before earthquake is definitely necessary.

## 7. Conclusions

In the years 2013 to 2017 the measurements of tectonic activity of ŚU provided necessary information to execute comparative study of the time distribution of seismic activity of FSM and to tectonic activity of ŚU. These analysis lead to the conclusion that the strong seismic shocks (magnitude ≥ 3.6) in FSM occur only in particular states of ŚU kinematic activity. We observed that phases of low velocity of deformations of the ŚU massif were necessary requirement for seismic events occurrence (see concentration close to zero of the values of amplitude precedents in the range of −22,000 to 22,000 μm/h Figure 7B, Figure 8B, Figure 9B, Figure 10B and Figure 11B and small values of the second derivatives in Table 3, at the moment of seismic event. The second requirement for seismic events occurrence is fulfillment of one of time precedents which was defined in Table 1 and presented in Figure 7A, Figure 8A, Figure 9A, Figure 10A and Figure 11A.

Finding of the time interdependence of kinematic activity of ŚU and seismic activity of FSM confirmed correctness of the thesis about the existence of large-scale, homogeneous field of tectonic forces, which was discuss in Section 1.

Following this reasoning large scale field of tectonic forces is subject to similar in the time variations in both geological units FSM and ŚU.

Therefore, during the phase of low-velocities of deformation in ŚU we ought to expect also the phase of low-velocities of deformation in the FSM. An analogous relationship between the phase of high-velocities of deformation of ŚU indicate existence of the phase of high-velocities of deformation of the FSM unit. As was previously shown the epochs of low-velocities of deformation of ŚU correspond the period of seismic activity in the FSM while the epochs of high-velocities of deformation of ŚU indicate aseismic period in FSM. In order to explain these interdependencies, we propose the following reasoning: Process of tectonic deformation of ŚU consist of two alternating epochs, namely periods of high velocities of deformation as well as periods of low velocities of deformation. During the periods of low- velocities of deformation, the VTAF cross zero level and vectors of deformations change their turn (Figure 5). At this time, we should expect that took place the change of turn of the field of tectonic forces which is the reason of velocity field variations.Moment of the change of turn of the field of tectonic forces results in a temporary stop of deformation of the orogen for several to several tens of hours (Figure 5). The temporary stop of the orogen deformation follow the final revers of direction of deformation and passage of the orogen through the state of extension.

Repeated coincidence between states of the low velocity of deformation (extension) and earthquakes suggest particularly important role of the epoch of extension in the process of adaptation of the orogen to the state of seismicity.

During the phase of extension in the rocky mass accrue process of rarefaction of the rocky medium, formation of free voids and discontinuities resulting creation additional space in the rocky medium.

Additional space is indispensable for absorption of the growth of rocks volume after seismic event. The laboratory experiences confirm that the volume of crumbled rocks is greater than volume of solid rocks before crush.

Following this reasoning the epoch of extension cause that the orogen becomes susceptible to be crushed. On the other hand, during the epoch of high speed of deformations (the compression phase) deficiency of free space in the orogen result lack of the seismic events.

The long-lasting observations from long water-tubes (>10 years) confirmed that strong earthquakes in FSM never happened while the epoch of high speed of deformations i.e., during the compression.

The seismic events which are discussed in this paper are associated with superimpose in the areas of mining exploitation of the local field of stresses with the large-scale field of stresses produced by activity of the north-central part of the Atlantic Rift.

In the blocks of rocks, where cumulated stresses of both mentioned sources exceed the mechanical strength of the rocks, occur destruction of material—the seismic event.

Existence of the time relations between earthquakes and epoch of low-velocities of deformation of ŚU i.e., epochs of extension show that the large-scale field of tectonic forces decide about susceptibility of the FSM rocky medium to destruction.

Therefore, large-scale field decide about possibility of seismic event occurrence while local field of stress of the anthropogenic genesis determines the location of shock only. Thus, it is impossible to determine the location of the seismic shock on the basis of observations of deformations of ŚU massif which is affected by large-scale field of tectonic forces only.

As was reported in the Section 6, certain number of epochs of extension proceed without seismic events. This observation suggest that epoch of extension and the correct implementation of time and amplitude precedents are necessary but not sufficient conditions to seismic event triggering (see for example situation in 14 December 2017; Figure 5). The reason of this is lack in presented discussion information of local stresses in mining area.

However, the periods of extension i.e., low-velocities of deformations are epochs of increasing of the seismic hazard and should be treated by mining workers as dangerous time. Therefore, determination in real time of periods of extension i.e., periods of susceptibility of the orogen to destruction can be used for construction tens of hours long extrapolation forecast of the seismic hazard in the FSM. Furthermore, the epochs of susceptibility of the orogen to destruction are the best moments for application procedure of effective relaxation of stresses in the orogen of mining area.

Practical utilization of presented method for assessment of increase of seismic hazard and determination of susceptibility of the orogen for destruction will be possible after modernization WTs to provide this information in the real time. To do it, all analogous modules in WTs should be replaced by digital measurement system.

Therefore, presented method of determination of derivatives of tectonic activity functions (VTAF) and evaluation of seismic activity in FSM is not a method of earthquakes prediction. On the basis of observations from the period 2013–2017 and longer we know for sure that the epochs of high-velocities of deformations of ŚU are aseismic in FSM. In the case of low-velocities of deformations i.e., during seismic epochs in FSM the case is not clear cut.

## Figures and Tables

**Figure 1 sensors-21-01603-f001:**
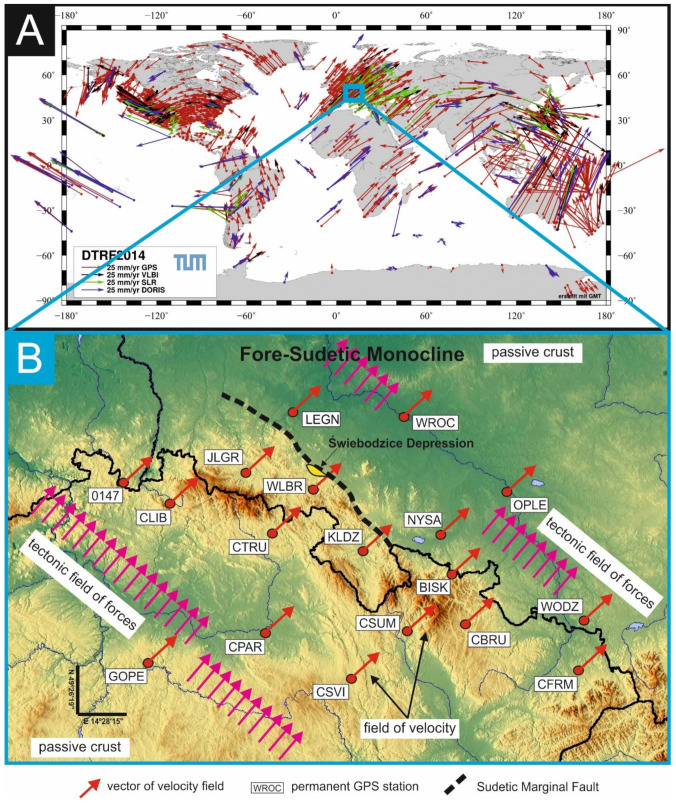
(**A**) DTRF2014 global horizontal velocity field [15]. (**B**) Map of the horizontal components of the velocity field based on time series of horizontal components of velocity vectors of the GPS stations in the ITRF2008 reference system [10,16] and expected direction of the field of tectonic force covering both Czech and Polish areas.

**Figure 2 sensors-21-01603-f002:**
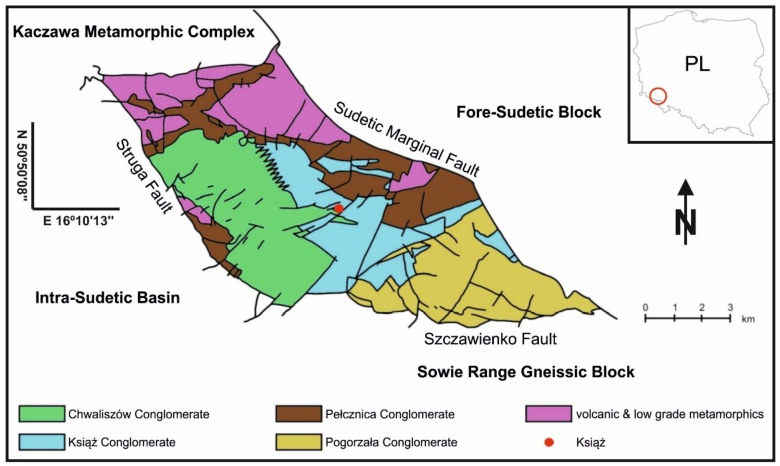
Simplified geological map of the Świebodzice Depression unit [25].

**Figure 3 sensors-21-01603-f003:**
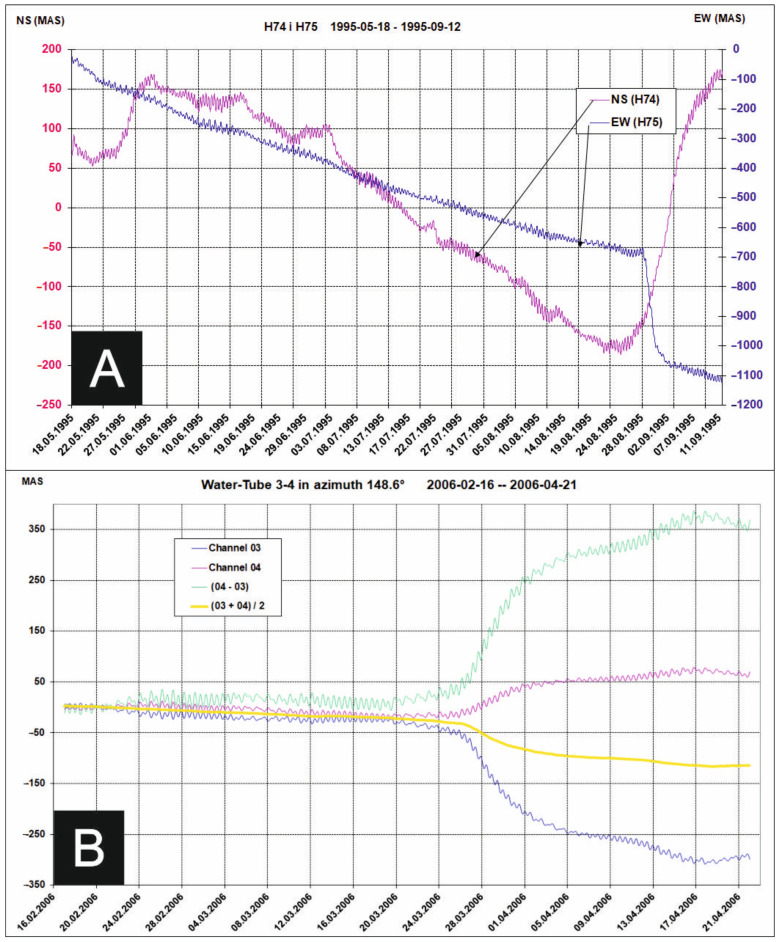
(**A**) Plots of tilting of foundation observed by HPs at the moment of large variations of equilibrium positions of pendulums caused by tectonic pressure pulses in August 1995. (**B**) effect of rapid water level variations on opposite channels (WT03-WT04) in March 2006 caused by effect of vertical motions of foundation. On the both figures are visible tidal undulations which provide reference to the scale of tectonic signals.

**Figure 4 sensors-21-01603-f004:**
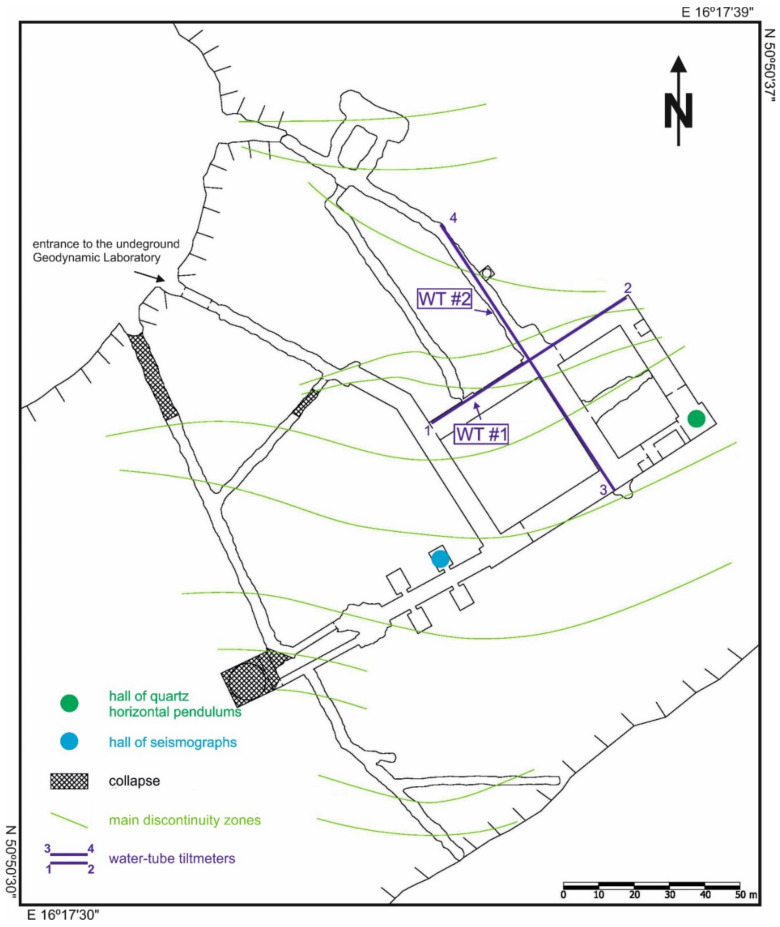
Plan of the underground of Książ Geodynamic Laboratory.

**Figure 5 sensors-21-01603-f005:**
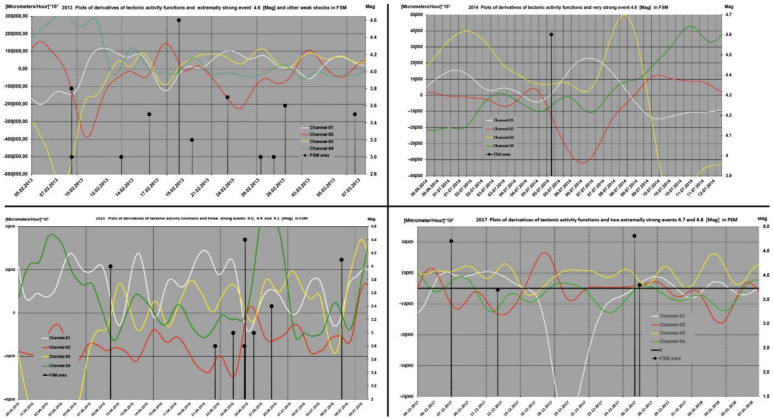
Example of exceptionally strong earthquakes in context of the variability of velocities of deformations i.e., derivatives of tectonic activity functions VTAF in 2013 (**top**-**left**), 2014 (**top**-**right**), 2015 (**bottom**-**left**) and 2017 (**bottom**-**right**). X axis: datum (DD-MM-YYY); Y-left axis: VTAF value [Micrometer/Hour] × 10^3^; Y-right axis: magnitude.

**Figure 6 sensors-21-01603-f006:**
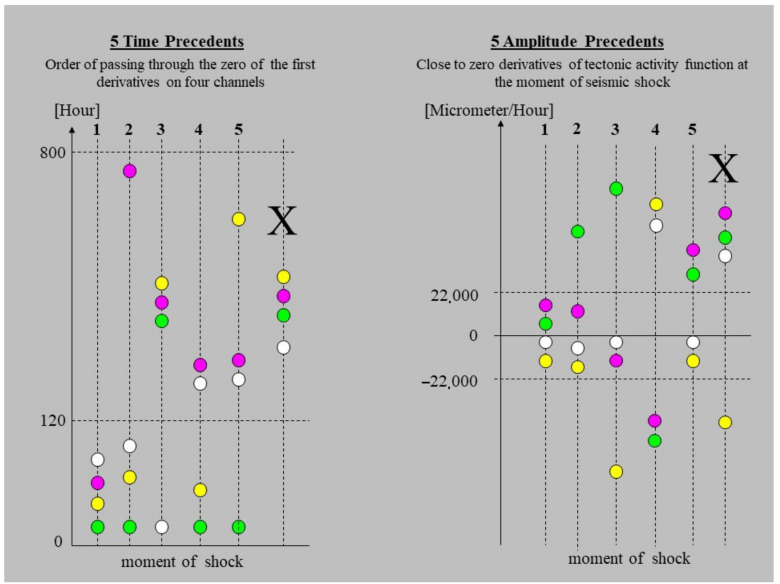
Five time and five amplitude precedents describing the process of deformation of the ŚU orogen which correspond to the state of seismicity of the FSM. Colored dots represent four VTAF. Vertical line marked by large “X” symbol shows configuration which never occur. “Moment of shock” refers to time of the earthquake occurrence.

**Figure 7 sensors-21-01603-f007:**
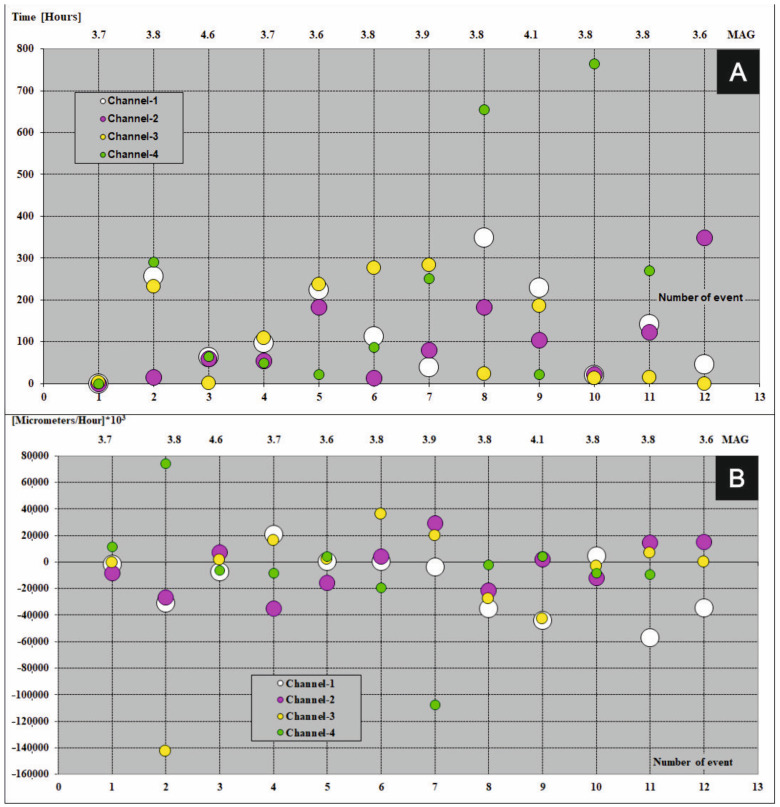
Time (**A**) and amplitude (**B**) precedents at the moments of seismic events (magnitude ≥ 3.6) in the year 2013. Colored dots represent four VTAF. Y axis: time (in hours for time precedents) and amplitude ([micrometers/hour] × 10^3^ for amplitude precedents); X-bottom axis: number of event; X-top axis: magnitude.

**Figure 8 sensors-21-01603-f008:**
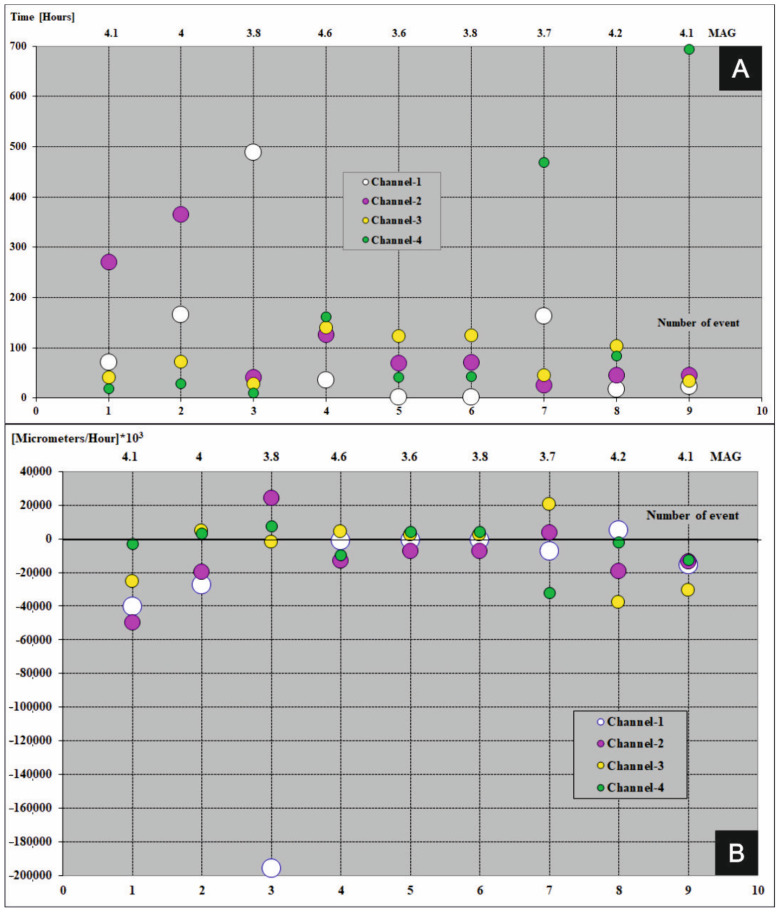
Time (**A**) and amplitude (**B**) precedents at the moments of seismic events (magnitude ≥ 3.6) in the year 2014. Colored dots represent four VTAF. Y axis: time (in hours for time precedents) and amplitude ([micrometers/hour] × 10^3^ for amplitude precedents); X-bottom axis: number of event; X-top axis: magnitude.

**Figure 9 sensors-21-01603-f009:**
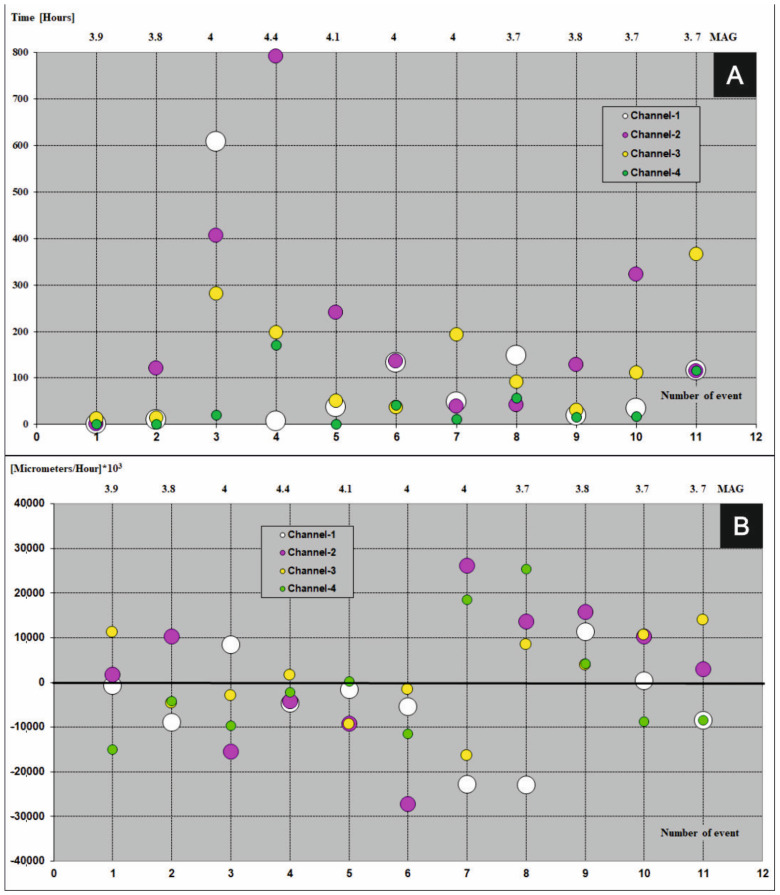
Time (**A**) and amplitude (**B**) precedents at the moments of seismic events (magnitude ≥ 3.6) in the year 2015. Colored dots represent four VTAF. Y axis: time (in hours for time precedents) and amplitude ([micrometers/hour] × 10^3^ for amplitude precedents); X-bottom axis: number of event; X-top axis: magnitude.

**Figure 10 sensors-21-01603-f010:**
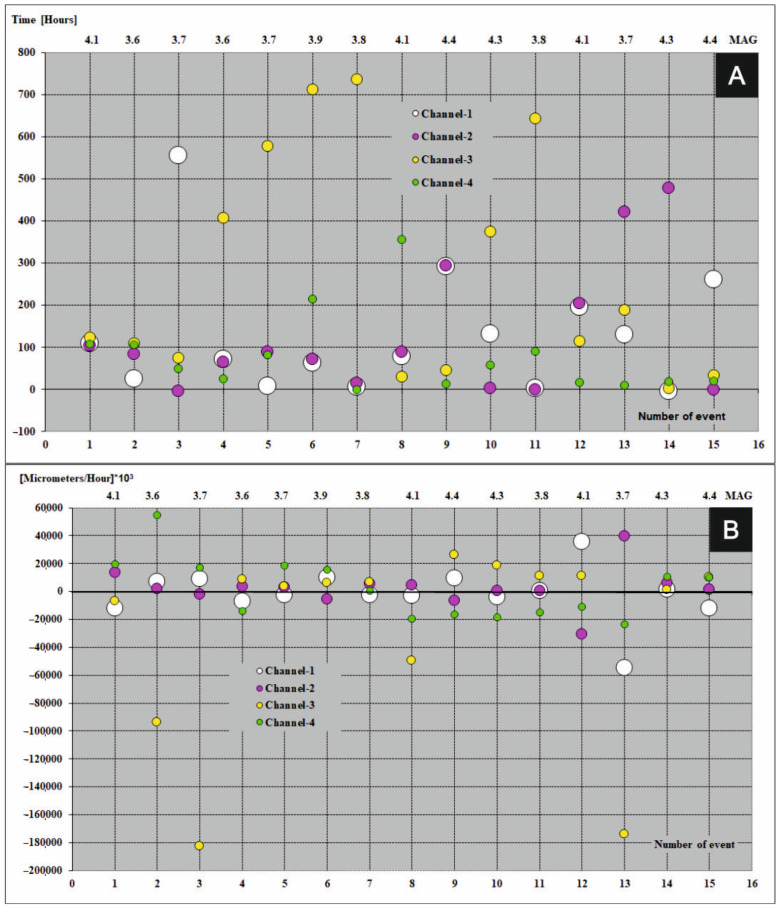
Time (**A**) and amplitude (**B**) precedents at the moments of seismic events (magnitude ≥ 3.6) in the year 2016. Colored dots represent four VTAF. Y axis: time (in hours for time precedents) and amplitude ([micrometers/hour] × 10^3^ for amplitude precedents); X-bottom axis: number of event; X-top axis: magnitude.

**Figure 11 sensors-21-01603-f011:**
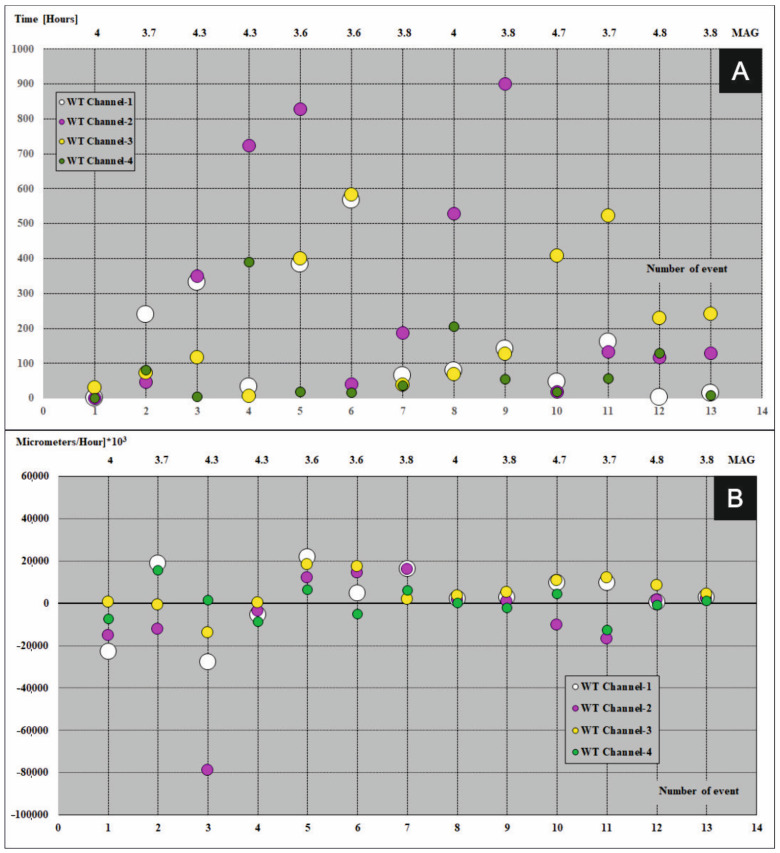
Time (**A**) and amplitude (**B**) precedents at the moments of seismic events (magnitude ≥ 3.6) in the year 2017. Colored dots represent four VTAF. Y axis: time (in hours for time precedents) and amplitude ([micrometers/hour] × 10^3^ for amplitude precedents); X-bottom axis: number of event; X-top axis: magnitude.

**Table 1 sensors-21-01603-t001:** Description of the time precedents T, as shown in Figure 6.

No.	Name of Time Precedent	Description
1	T-1	4 passages through zero of the VTAF within ca. 120 h before the shock (very strong condition, very highly probable seismic event during the next 100 h after last passage)
2	T-2	3 passages through zero of the VTAF within ca. 120 h before the shock and preceding the passage up to 700 h before the shock (strong condition, seismic shock highly probable during the next 100 h after last passage)
3	T-3	1 passage through zero of the VTAF within ca. 50 h before the shock and 3 VTAF passages through zero 300 h before the shock (medium-strong condition, shock is expected during the next 50 h)
4	T-4	2 passages through zero of the VTAF within ca. 100 h before the shock and 2 passages close to each other preceded—up to 300 h before the shock (medium-strong condition, shock is expected within 100 h after last passage)
5	T-5	2 passages through zero of the VTAF with channels close in time to each other within ca. 100 h. Two other channels are shifted in time less than <600 h—shock is expected within ca. 100 h after last passage

**Table 2 sensors-21-01603-t002:** Description of the amplitude precedents A, as shown in Figure 6.

No.	Name of Amplitude Precedent	Description
1	A-1	Four-fold concentration of the VTAF and close to zero (values of derivatives from the range –22,000 to 22,000 μm/h)—very strong condition –seismic event expected ca. 100 h after last passage of the VTAF through zero
2	A-2	Three-fold concentration of the VTAF—two cases a and b: a: 3 channels close to zero (values of derivatives from –22,000 to 22,000 μm/h) and 1 VTAF channel distant which value is <80,000 μm/h b: 1 channel close to zero (values of VTAF from –22,000 to 22,000 μm/h) and other channels concentrated in distant >22,000 μm/h from zero – seismic event expected ca. 100 h after last passage of the VTAF through zero
3	A-3	Two-fold concentration of the VTAF close to zero (values of derivatives from –22,000 to 22,000 μm/h) and two channels with mutually symmetrical values at the distance of <80,000 μm/h. Seismic event expected ca. 100 h after last passage of the VTAF through zero)
4	A-4	All of the VTAF values distant to zero and of double symmetry of channels. This strong condition occurred rare; high probability of seismic event within 100 h after last passage of the VTAF through zero)
5	A-5	Two-fold concentration of the VTAF close to zero (values of VTAF from the range –25,000 to 25,000 μm/h) and two-fold concentration of the VTAF distant from zero, seismic event expected <100 h after last passage of the VTAF through zero.

**Table 3 sensors-21-01603-t003:** Combination of seismic events which occurred in the FSM in the years 2013–2017 with values of VTAF, time lags of zero crossing of the VTAF as well as time and amplitude precedents.

Year	No	Datum (YYYY-MM-DD) and Time (hours, UT)	Mag [MJ]	WT1 Channel 1		WT1 Channel 2		WT2 Channel 3		WT2 Channel 4		Precedents	
				VTAF	Time Lag [hours]	VTAF	Time Lag [hours]	VTAF	Time Lag [hours]	VTAF	Time Lag [hours]	T	A
2013	1	2013-02-09 17	3.7	−1446	0	−8464	0	−821	2	11,304	0	T-1	A-1
	2	2013-03-09 23	3.8	−30,839	256	−26,482	15	−142,464	232	73,946	291	T-3	A-3
	3	2013-03-19 21	4.6	−7554	62	7429	59	1304	1	−6089	65	T-1	A-1
	4	2013-03-24 09	3.7	20,929	96	−35,018	54	16,268	109	−8196	49	T-1	A-2a
	5	2013-03-29 17	3.6	411	224	−15,571	182	2268	237	4375	22	T-3	A-1
	6	2013-05-27 21	3.8	571	113	4321	13	36,196	277	−19,643	86	T-2	A-3
	7	2013-07-13 04	3.9	−3893	38	29,393	80	19,946	283	−107,679	251	T-4	A-2a
	8	2013-07-29 23	3.8	−34,857	348	−21,714	183	−27,589	24	−2196	654	T-3	A-2b
	9	2013-09-20 01	4.1	−43,768	229	1929	104	−42,804	185	4286	21	T-4	A-5
	10	2013-11-23 23	3.8	4554	19	−12,125	21	−3339	13	−8179	765	T-2	A-1
	11	2013-12-10 06	3.8	−56,786	142	14,339	122	6768	14	−9625	269	T-3	A-2a
2014	1	2014-03-16 06	4.1	−40,089	71	−49,589	270	−25,321	41	−2804	18	T-2	A-2b
	2	2014-03-20 05	4.0	−27,554	166	−19,839	365	4518	72	3179	29	T-5	A-5
	3	2014-04-02 15	3.8	−196,054	488	24,071	41	−2196	29	7411	10	T-2	A-3
	4	2014-07-05 12	4.6	−1321	35	−12,982	126	4268	140	−9554	161	T-3	A-1
	5	2014-07-21 16	3.6	−464	1	−7464	70	1982	124	4321	42	T-2	A-1
	6	2014-07-21 16	3.8	−464	2	−7464	71	1982	125	4321	43	T-2	A-1
	7	2014-08-20 04	3.7	−7304	163	3875	26	20,179	45	−32,214	469	T-5	A-3
	8	2014-10-26 03	4.2	5179	17	−19,339	45	−38,054	104	−1929	84	T-1	A-2a
	9	2014-12-12 02	4.1	−15,679	23	−13,321	46	−30,643	34	−12,607	694	T-1	A-2a
2015	1	2015-02-05 04	3.9	−600	0	1850	0	11,300	13	−14,950	0	T-1	A-3
	2	2015-02-12 18	3.8	−8850	10	10,300	120	−4600	14	−4150	0	T-1	A-1
	3	2015-06-22 04	4.0	8500	607	−15,435	406	−2950	282	−9700	20	T-3	A-1
	4	2015-07-08 06	4.4	−4550	6	−4135	792	1700	198	−2150	170	T-3	A-1
	5	2015-07-19 19	4.1	−1650	37	−9135	241	−9300	50	300	0	T-2	A-1
	6	2015-09-09 19	4.0	−5400	133	−27,185	136	−1550	37	−11,450	41	T-4	A-2a
	7	2015-10-29 02	4.0	−22,700	47	26,165	39	−16,400	193	18,550	11	T-2	A-4
	8	2015-11-20 07	3.7	−22,950	147	13,715	41	8500	92	25,450	57	T-2	A-3
2016	1	2016-02-25 04	4.1	−7119	72	3476	64	8357	407	−14,161	25	T-2	A-1
	2	2016-03-08 03	3.6	−2667	7	3238	89	3524	578	18,750	81	T-2	A-1
	3	2016-04-08 15	3.7	10,357	63	−5262	71	5881	711	15,839	214	T-5	A-1
	4	2016-04-28 12	3.6	−2286	5	5786	14	6667	736	768	0	T-2	A-1
	5	2016-05-05 15	3.7	−2833	78	4571	88	−49,548	30	−19,482	356	T-2	A-2a
	6	2016-05-11 04	3.9	9595	291	−6214	294	26,000	45	−16,464	12	T-4	A-2a
	7	2016-05-12 05	3.8	−3929	132	405	2	18,762	374	−18,268	58	T-5	A-1
	8	2016-06-02 04	4.1	548	2	714	169	11,357	643	−14,911	90	T-5	A1
	9	2016-07-30 19	4.4	35,690	196	−30,500	204	11,000	115	−10,929	16	T-3	A-3
	10	2016-08-13 12	4.3	−54,405	130	39,619	421	−174,214	189	−23,393	9	T-3	*LD
	11	2016-08-24 17	3.8	1429	7	5857	477	1143	2	10,750	18	T-2	A-1
	12	2016-09-14 07	4.1	−11,762	260	1571	0	9905	34	10,196	20	T-2	A-1
	13	2016-10-15 15	3.7	−7119	72	3476	64	8357	407	−14,161	25	T-2	A-1
	14	2016-10-17 23	4.3	−2667	7	3238	89	3524	578	18,750	81	T-2	A-1
	15	2016-11-29 20	4.4	10,357	63	−5262	71	5881	711	15,839	214	T-5	A-1
2017	1	2017-01-22 19	4.0	−22,619	0	−15,357	0	500	30	−7357	0	T-1	A-2a
	2	2017-03-17 08	3.7	18,881	240	−12,238	46	−881	72	15,595	81	T-2	A-1
	3	2017-04-08 22	4.3	−27,667	331	−78,905	349	−13,833	116	1524	5	T-4	A2a/A5
	4	2017-05-31 20	4.3	−5405	33	−3643	723	262	6	−8595	391	T-5	A-1
	5	2017-10-13 13	3.6	22,048	385	12,190	827	18,262	400	6429	18	T-3	A-2a
	6	2017-10-21 03	3.6	4667	567	14,381	41	17,429	582	−5190	16	T-4	A-1
	7	2017-10-27 05	3.8	16,405	65	15,833	187	1714	39	6048	36	T-2	A-1
	8	2017-11-10 11	4.0	2048	79	1452	529	3452	68	71	205	T-4	A-1
	9	2017-11-25 23	3.8	2810	141	786	901	5167	127	−2024	54	T-2	A-1
	10	2017-12-07 17	4.7	9619	47	−10,143	18	10,881	409	4643	18	T-2	A-1
	11	2017-12-12 11	3.7	9667	161	−16,762	132	12,095	523	−12,667	57	T-3	A-1
	12	2017-12-26 11	4.8	524	3	1381	117	8524	230	−857	129	T-2	A-1
	13	2017-12-26 23	3.8	2786	15	2048	129	4595	242	1143	8	T-2	A-1

* LD—lack of definition.

**Table 4 sensors-21-01603-t004:** Distribution of the amplitude and time precedents in relation to seismic events energy in the years 2013–2017.

Mag		2013	2014	2015	2016	2017	Sum of Events	Amplitude Precedents						Time Precedents				
								A1	A2a	A2b	A3	A4	A5	T1	T2	T3	T4	T5
3.6/3.7	**Sum of Events**	3	2	1	5	4	15											
	**Precedent A**	A1, A2a, A1	A1, A3	A3	A3, A2a, A1, A1, *LD	A1, A2a, A1, A1		8	3		3							
	**Precedent T**	T1, T1, T3	T2, T5	T2	T1, T2, T2, T2, T3	T2, T3, T4, T3								3	6	4	1	1
3.8/3.9	**Num. of Event**	6	2	3	3	3	17											
	**Precedent A**	A3, A3, A2a, A2b, A1, A2a	A3, A1	A3, A1, A1	A1, A1, A1	A1, A1, A1		10	2	1	4							
	**Precedent T**	T3, T2, T4, T3, T2, T3	T2, T2	T1, T1, T2	T5, T2, T5	T2, T2, T2								2	9	3	1	2
4.0/4.1	**Num. of Events**	1	3	4	3	2	13											
	**Precedent A**	A5	A2b, A5, A2a	A1, A1, A2a, A4	A1, A2a, A3	A2a, A1		4	4	1	1	1	2					
	**Precedent T**	T4	T2, T5, T1	T3, T2, T4, T2	T1, T2, T3	T1, T4								3	4	2	3	1
4.2/4.3	**Num. of Events**	-	1	-	2	2	5											
	**Precedent A**	-	A2a	-	A1, A1	A2a/A5, A1		3	2									
	**Precedent T**	-	T1	-	T5, T2	T4, T5								1	1		1	2
4.4/4.5	**Num. of Events**	-	-	1	2	-	3											
	**Precedent A**	-	-	A1	A2a, A1	-		2	1									
	**Precedent T**	-	-	T3	T4, T2	-									1	1	1	
4.6/4.7	**Num. of Events**	1	1	-	-	1	3											
	**Precedent A**	A1	A1	-	-	A1		3										
	**Precedent T**	T1	T3	-	-	T2								1	1	1		
4.8/4.9	**Num. of Events**	-	-	-	-	1	1											
	**Precedent A**	-	-	-	-	A1		1										
	**Precedent T**	-	-	-	-	T2									1			
**Σ of Events**		11	9	9	15	13	**57**	**Sum of Amplitude Precedents in Category**						**Sum of Time Precedents in Category**				
**Σ of Precedents**								**31**	**12**	**2**	**8**	**1**	**2**	**10**	**23**	**11**	**7**	**6**

* LD—lack of definition.

## Data Availability

The data presented in this study are available on request from the corresponding author. The data are not publicly available due to privacy restrictions.

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
