# Peer review of "Time Distribution of Strong Seismic Events in the Fore-Sudetic Monocline in Context of Signals Registered by Water-Tube Gauges in Książ Geodynamic Laboratory"

_sensors, 2021, doi:10.3390/s21051603_

Round 1

Reviewer 1 Report

The study presented in this paper shows signals of kinematic activity, as recorded using water tube tiltmeters in the Swiebodzice Depression, that correlate with seismic events in the Fore-Sudetic Monocline.

This study has potential to be a significant contribution to understanding regional tectonic kinematics related to seismic activity. However, the delivery of the paper prevents thorough understanding of the study, and the presentation of results lacks thorough enough context to make evaluation of the methods possible.

An extensive amount of English editing is necessary to make this paper understandable. In the bulleted list below, I have highlighted a few parts that I really cannot understand, but the necessary editing needs to be accomplished throughout. Style, grammar, and in some cases word and terminology use needs to be checked and corrected.

The main concern I have about the research methods and the conclusions of the study is that the main results, shown in Figure 7, are not set in context of what the norm is. It is impossible for the reader to assess whether the results discussed are meaningful or significant. For example, one of the conditions discussed as correlating with seismic events is values of VTAF between +-22,000 um/h, but as far as I could tell, neither the full range of values of the VTAF, nor the typical values are presented. Figure 5 looks like it is presenting a full time series of the VTAF values, which is part of what is needed, but the text is so small that the graph is illegible. Neither the units of the x-axis, nor the total range (time span?) covered in the plot is indicated.

Specific comments

The introduction does ok with setting the framework of the underlying assumptions, but it is lacking a broader context, as well as context for the questions being investigated. The introduction starts right off with mentioning the FSM and SU, but does not first describe where they are and why they matter in a broader context.

Line 38 "many years long observations of the Earth" is too vague a statement. Observations of what aspects/ processes of the earth?

Line 45 "e.g. areas of continental plates collisions" does not make sense given the first part of the sentence that says that "areas of rifts, subduction zones" are among the exceptions. Rifts and subduction zones do not occur only where continental plates collide. Also, "other oddities" is vague and doesn't communicate much about the exceptions to the norm.

Line 70 I do not know what you mean by "tectonic pulses". I don't think you explain what TM-71 crack gauges are.

Line 73 When you say "route of the SFM" do you mean "strike of the SFM"?

Line 95 What type of "measure gauges" are you referring to? What exactly is being measured?

Line 101 Do you mean "until a depth of 10 km or more"?

Lines 101-102 What do you mean by "contact of the surface rocks of the SU massif with deep stratum of Earth's crust"?

Line 119 Maybe "blocks of the fault" is better wording than "wings".

Line 120 It would be good to say here what specific instrument are you using to measure the radon concentrations.

Line 128 What are WTs? I think you missed defining this abbreviation.

Lines 133-135 The sentence does not make sense. The wording is confusing.

Line 137 What is the "foundation" you are referring to? Is it where the WTs are mounted or something like that?

Figure 3 needs axis labels with units.

On Figure 5, labels on charts are too small to see without going to 200% zoom. X-axis needs units label. It would be helpful to indicate the full time range covered in each plot in the figure caption.

Line 207 "presented in Chapter 1" Do you mean section 1 of this paper?

Lines 216-219 introduces (if I understand right), the idea of the "precedents" that are discussed in the following section. For me, this needs to be set up better. What is meant by "precedents"? What is the significance?

Figure 6: The label "The characteristic which not exist" does not make sense to me. First, the grammar is not right, and second, it is unclear what is referred to by it. Does the white X mark a specific characteristic that doesn't exist? If so, it may be more clear to indicate that in the figure caption. What is meant by "Moment of shock"? That label appears in both the time and amplitude plots, so it's unclear what it is referring to, time or size?

Tables 1 and 2 need more space between each line in the Description column. As it is, it is very difficult to see what belongs to each row.

Table 1, No. 3. This sentence doesn't make sense.

Line 252: What is the typical range of values? How different is the +-22,000 range from the typical range?

Line 253: What is meant by "double symmetry"? I do not see where this case/term is described.

Figure 7 most of the text on the charts is too small to see. If all the color legends are the same, you could have just one larger legend for the whole figure or one for each of time and amplitude. It is not clear to me what relationships are being shown in time. Are the colored dots the recordings or VTAF values at the time of each earthquake? Please elaborate more in the figure caption, as well as clarifying in the text.

Lines 306-307 Here a magnitude 3.7 earthquake is described as "weak", wherease throughout the rest of the paper, earthquakes of magnitude 3.6 and above are described as "strong". What is different about this "weak" magnitude 3.7 earthquake that leads to the conclusion that it is anthropogenic?

Lines 365-366: There are numerous studies on characteristics of anthropogenic seismicity. How do the findings in this study relate to what is already know of characteristics of anthropogenic seismicity? This will help establish if this is a reasonable conclusion and show what these findings add to existing knowledge about anthropogenic seismicity.

Lines 383-385 It needs to be explained better why a "slowing-down of kinematic activity" is indicated by the observations.

Line 411 I need to know what the typical range of these values is, so I could see how significant these so called "low" values are. The figures show only the values that occur a the times of the earthquakes, so I do not have a way to see how the patterns highlighted in the study relate to background or typical patterns. Maybe this is giving in Figure 5, but it is unclear.

Line 417: "Chapter 1" do you mean section 1 of this paper?

Lines 430-431 what is meant by the phrase "change their turn"?

Lines 434-438 Are there examples in the literature of fields of tectonic forces changing on a similar scale of "several tens of hours" described in other places?

Lines 446-447 I don't see where this claim is demonstrated in the results.

Line 449 I'm not certain "destruction" is the best word here. I'm not sure what you are meaning.

Lines 458-459 the wording in this sentence does not make sense.

Lines 465-467 Good, this is an important point to make.

Lines 472-471 The wording in this sentence is confusing and seems incomplete.

Lines 478-480 This sentence doesn't make sense.

Author Response

Dear Reviewer, thank you for your comments. We have revised our manuscript and applied the necessary corrections. Please check our answers in the attached file as well as the manuscript – all changes were marked by the red color of the font for better recognition. We have also checked/done English editing thanks to the MDPI service

Reviewer 2 Report

Dearest Editorial Office,
I finished the review work about the manuscript sensors-1073127 entitled: “Time distribution of strong
seismic events in the Fore-Sudetic Monocline in contexts of signals registered by water-tube gauges in Książ
Geodynamic Laboratory” by Marek Kaczorowski, Damian Kasza, Ryszard Zdunek and Roman Wronowski.

The paper is rich in analysis so suitable for this journal but with several revisions.
General considerations and suggestions to improve this work
(1) The Fig. 1b is not georeferenced.
(2) Probably it is necessary a Fig. 1c with seismicity used in the analysis.
(3) The manuscript is submitted to “Sensors” but probably is necessary a big quantity of information about
your sensors, like technical characteristics, sensitivity, figures, photos, how do you acquire the signals ?,
sampling and so on.
(4) The Fig. 2 is not georeferenced.
(5) About Fig. 3: more details about the units of measure; it is not clear the presence of channel 03 or 04 with
respect to NS (H74) or EW (H75).
(6) Why in the Fig. 3 you plotted only 4 months of 1995 and only 2 months of 2006 ?
(7) The Fig. 5 is very bad like resolution, the fonts are very small, the legends are not clear, please more detail
about the considered earthquakes and to to extend the caption.
(8) In fig. 6 the choice of black background is not ideal and it is necessary to extend the caption.
(9) The Fig. 7 is very bad like resolution, the fonts are very small, the legends are not clear and to to extend
the caption.
In the Table 3:
(a) it is necessary to extend the caption.
(b) To define “time leg”, probably “time lag” ?
(c) How do you explain several hundred hours of time lags ?
(d) For each earthquake it is necessary not only date bat also time in UT.
(e) Please more details about earthquake catalogues.
(f) Probably an important information it is the epicentral distance for each earthquake.
(g) Also azimuth and focal mechanism information for each earthquake could be important.
(h) In the fig. 4 I noted a presence of the “hall of seismographs”; do you used this data ?

Author Response

(The authors gave the same response as above.)

Round 2

Reviewer 2 Report

I am satisfied with the authors' answers